# A Novel Structure of Rubber Ring for Hydraulic Buffer Seal Based on Numerical Simulation

Lin Zhang  and Xiaohui Wei *

Key Laboratory of Fundamental Science for National Defense-Advanced Design Technology of Flight Vehicle, Nanjing University of Aeronautics and Astronautics, Nanjing 210016, China; lin_zhang@nuaa.edu.cn
* wei_xiaohui@nuaa.edu.cn; Tel.: +86-1569-957-5280

**Featured Application: 1. A novel structure of a rubber ring with butterfly geometry is delivered based on numerical simulation, and proved to be better than current used O-ring. 2. The effects of pre-compression rate and hydraulic pressure on the static sealing performance of a rubber ring are obtained. 3. The simulation method of installation boundary for the sealing analysis of a rubber ring is discussed, and a simulation method suitable for rubber ring seal analysis is proposed.**

**Abstract:** Landing gear is a key load-bearing structure of aircraft during ground operation, and the landing capacity of landing gear is determined by the performance of buffer. To solve the problem of buffer failure caused by insufficient static sealing of a rubber ring at groove side, a new structure of a butterfly rubber ring is proposed by analyzing the factors affecting sealing performance of the rubber ring. First, the constitutive equation of rubber material is derived based on the theory of hyper-elasticity, and the material parameters are obtained by fitting the experimental data. Then, by analyzing the simulation method of installation mode and installation stroke, the simulation method suitable for calculating the sealing performance of the rubber ring is established. The linear fitting formulas with Pearson coefficient greater than 0.92 are used to discuss the influence of pre-compression rate and hydraulic pressure on the sealing performance of the rubber ring. Compared with O-ring, the contact pressure of butterfly-ring is increased by 30% in assembly state and 14% in working state. The results show that the butterfly rubber ring has excellent static sealing performance. It is concluded that improving the configuration of the sealing ring can solve the insufficient unilateral sealing of the hydraulic buffer.

**Keywords:** rubber ring; contact pressure; hydraulic buffer; installation mode; numerical simulation



## 1. Introduction

Landing buffer capability of landing gear buffer is the key to evaluate landing gear performance, especially for carrier-based fighter and hypersonic aircraft [1]. After buffering the ground load by the buffer, the damage caused by the ground load on the landing gear can be reduced, and the service life of the landing gear can be improved. The basic requirement of buffer design is good sealing performance [2]. Rubber material has hyperelastic performance, which is widely used in the sealing structure of buffer. As the main seal of hydraulic buffer, the sealing performance of O-ring is directly related to the working performance of buffer [3]. The research on the factors affecting the sealing performance of the rubber ring is helpful to calculate the sealing performance of the rubber ring and improve the accuracy of buffer design.

Due to the nonlinear behavior of material, geometry and contact, the finite element analysis (FEA) is widely used as a research method of rubber ring sealing performance [4]. As a kind of hyperelastic material, the constitutive equations commonly used in numerical calculation are Arruda Boyce, van der Waals, Ogden, Mooney–Rivlin (M-R), etc. Ali

reviewed the widely used constitutive models of rubber materials in FEA, and pointed out that M-R model is suitable for small deformation and medium deformation [5]. Huang pointed out that the seal structure is axisymmetric, so the two-dimensional axisymmetric model can be used for the numerical calculation of the seal ring. Also, the two-dimensional axisymmetric model of the seal structure was established by using ANSYS software, and the validity of the two-dimensional axisymmetric numerical model is verified by the FEA of the sealing surface and the back support structure of the roller bit bearing seal [6,7]. Zhou established five contact pairs and analyzed the state of contact surface with the form of pseudo element. By comparing the numerical results with the experimental data, Zhou pointed out that it is feasible to simulate the contact pressure distribution of bearing seal structure through contact pairs [8]. The FEA method is an effective method to study the sealing performance of the rubber ring [9].

The most important factor in the design of hydraulic buffer seal structure is the configuration of the rubber ring. The configuration of the rubber ring determines the contact peak pressure and contact length, both of which are used as the evaluation index of seal performance. Angus Jean analyzed the influence of geometric shape on the deformation of rubber structure, and pointed out that different structural shapes have a great influence on the sealing performance of the rubber ring [10]. A lot of work has been done in the structural design of rubber sealing ring, and some new rubber sealing rings have been used in buffer seal, such as D-ring, T-ring, U-ring, X-ring, Y-ring, etc. The geometry of D-ring, which is designed to increase the contact length between groove and D-ring, could be seen as a half of O-ring. Zhou compared D-ring with O-ring in the sealing performance, and concluded the sealing performance of D-ring is better than O-ring from the perspective of contact peak stress. Also pointed the disadvantage of extrusion through buffer gap for D-ring [11]. Mose analyzed the extrusion of D-ring with hydraulic pressure 5.89 MPa, and concluded the extrusion is likely to cause sealing failure. The sealing disadvantage of D-ring is laid on stress concentration at corner under static sealing and extrusion through gap under dynamic sealing [12]. The geometric shape of X-ring is benefit for dynamic condition by decreasing friction. Shin showed the higher contact stress of X-ring than O-ring under certain research condition, but the ultimate pressure of extrusion is only 3.92 MPa for X-ring [13]. The disadvantage of X-ring is the contact length under low pressure, which is harmful to the sealing of the rubber ring [14]. Furthermore, the X-ring is not suitable for landing gear buffer, due to the limitation of hydraulic pressure. The sealing characteristic of Y-ring relied on the surface of lips to coupling, as a lip-type rubber ring. Cui analyzed the effect of hydraulic pressure on Y-ring sealing performance, and found out the location of the maximum stress and largest deformation area according to seal failure criterion [15]. The disadvantage of thresh tendency under high pressure is a weakness for Y-ring. The type of O-ring is widely used in hydraulic buffer of landing gear, due to the stability of contact length and pressure [16]. According to test data on 21 joint seals, Jahangir concluded that the properties of joint seals are differed by geometry shape and manufacture method [17]. As the seal of hydraulic buffer for landing gear, the disadvantage of lower contact pressure at groove side than piston rod side still needs to be modified for O-ring during assemble process [18].

To improve the sealing pressure at groove side of articulated landing gear buffer, the numerical model of rubber ring sealing analysis is discussed, the sealing characteristics of the rubber ring are analyzed, and the sealing performance of proposed butterfly-ring and original O-ring is compared. First, the stress–strain formula of the rubber ring is derived based on the theory of hyper-elasticity, and the material parameters of M-R are obtained by using compression test data from reference. Secondly, using nonlinear numerical technique, three typical simulation methods of installation mode and installation stroke are discussed. On this basis, the numerical model of rubber sealing ring is established, and 16 groups of 154 simulations are carried out. Based on the numerical test of O-ring, the relationship between the sealing characteristics and load of O-ring in installation mode and working

mode is discussed, including pre-compression rate and hydraulic pressure. And then, by comparing with O-ring, the sealing performance of butterfly-ring is discussed.

## 2. Materials and Structure

### 2.1. Material

The material used in this buffer seal is Nitrile rubber(NBR), which is widely used in landing gear buffer, considered the pressure resistance of NBR. Based on the assumption of hyper-elasticity theory, the deformation and stress of rubber-like material which is employed by the strain–energy density function, is written as a function of the strain invariant [19].

$$W = f(I_1, I_2, I_3) \tag{1}$$

where $W$ is the strain–energy function, $\lambda$ represents for the invariant of the Cauchy–Green strain tensor [20]. Furthermore, the invariants of the Cauchy–Green strain tensor could be expressed by the principal stretches ratios $\lambda_1, \lambda_2, \lambda_3$.

$$\begin{cases} I_1 &= \lambda_1^2 + \lambda_2^2 + \lambda_3^2 \\ I_2 &= (\lambda_1\lambda_2)^2 + (\lambda_2\lambda_3)^2 + (\lambda_3\lambda_1)^2 \\ I_3 &= (\lambda_1\lambda_2\lambda_3)^2 \end{cases} \tag{2}$$

In addition, the formula of the Cauchy stress tensor $\sigma^{ij}$ is expressed in terms of the principal stretches,

$$\begin{cases} \sigma_1 = 2\lambda_1 \left[ \frac{\partial W}{\partial I_1} + (\lambda_2^2 + \lambda_3^2)\frac{\partial W}{\partial I_2} + \lambda_2^2\lambda_3^2\frac{\partial W}{\partial I_3} \right] \\ \sigma_2 = 2\lambda_2 \left[ \frac{\partial W}{\partial I_1} + (\lambda_1^2 + \lambda_3^2)\frac{\partial W}{\partial I_2} + \lambda_1^2\lambda_3^2\frac{\partial W}{\partial I_3} \right] \\ \sigma_3 = 2\lambda_3 \left[ \frac{\partial W}{\partial I_1} + (\lambda_1^2 + \lambda_2^2)\frac{\partial W}{\partial I_2} + \lambda_1^2\lambda_2^2\frac{\partial W}{\partial I_3} \right] \end{cases} \tag{3}$$

The two parameter M-R model for approximately incompressible material is adopted,

$$W = C_{10}(I_1 - 3) + C_{01}(I_2 - 3) + \left( \frac{C_{10}}{2} + C_{01} \right)\left( \frac{1}{I_3^2} - 1 \right) + \frac{C_{10}(5\nu - 2) + C_{01}(11\nu - 5)}{2(1 - 2\nu)}(I_3 - 1)^2 \tag{4}$$

where $C_{10}$ and $C_{01}$ are the material parameters. As the rubber material is assumed to be approximately incompressible [21], i.e., the Poisson ratio $\nu = 0.499 \approx 0.5$, $I_3 = 1$. Therefore, the two parameter M-R model is written as,

$$W = C_{10}(I_1 - 3) + C_{01}(I_2 - 3) \tag{5}$$

In addition, the uniaxial compression tests,

$$\begin{cases} \lambda_1 = \frac{1}{\lambda_2^2} = \frac{1}{\lambda_3^2} = \lambda \\ \sigma_1 = 2\lambda_1 \left[ \frac{\partial W}{\partial I_1} + (\lambda_2^2 + \lambda_3^2)\frac{\partial W}{\partial I_2} \right] \\ \sigma_2 = \sigma_3 = 0 \end{cases} \tag{6}$$

where $\lambda$ is the strain value. By Equations (5) and (6), the Cauchy stress is developed as:

$$\sigma = \frac{2\lambda^4}{\lambda^3 - 1}C_{10} + \frac{2\lambda^3}{\lambda^3 - 1}C_{01} \tag{7}$$

According to the data of rubber compression experiment quoted from Reference [8], the material parameters of M-R are fitted as $C_{10} = 1.84$, $C_{01} = 0.47$ with Equation (7).

## 2.2. Structure

The hydraulic buffer is the key device of articulated landing gear which is used to bear and transmit the vertical load from ground to fuselage. As presented in Figure 1, the rubber ring is located at the bottom of the hydraulic buffer, which plays a sealing role to prevent the leakage of the hydraulic system. The structure of O-ring is adopted as the type of inner seal and outer seal for hydraulic buffer.

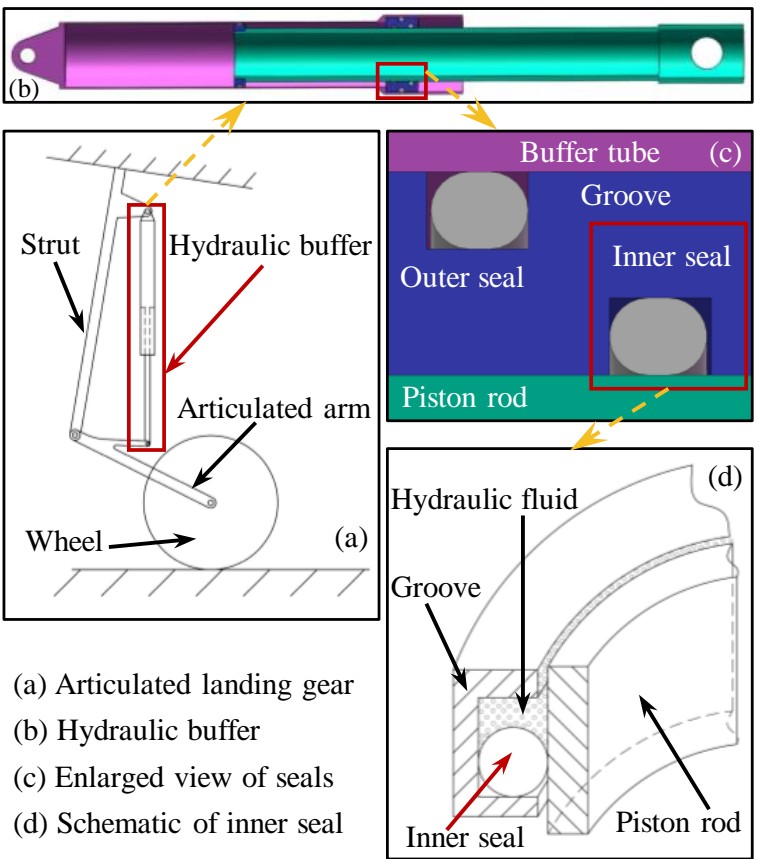

(a) Articulated landing gear

(b) Hydraulic buffer

(c) Enlarged view of seals

(d) Schematic of inner seal

**Figure 1.** Schematic of seals for articulated landing gear. The hydraulic buffer is a key structure of load-bearing in the articulated landing gear. And the hydraulic buffer of articulated landing gear is composed of buffer tube, groove, seals, and piston rod. To prevent the leakage of hydraulic fluid, the rubber O-rings are used as the inner seal and outer seal.

## 3. Modeling

First, the nonlinear finite element model of seal structure is established. Then, three typical installation modes are simulated to establish the suitable installation mode, and hydraulic pressure is applied to the deformed rubber ring. On this basis, the sealing performance of the original structure and the proposed structure are analyzed and compared.

### 3.1. Finite Element Model

According to the rubber ring of hydraulic buffer used in articulated landing gear, considered the axisymmetric characteristic of geometry and load condition [22,23], the axisymmetric model of rubber O-ring is established with section diameter 7 mm as shown in Figure 2a. Specific geometry size of groove and piston rod are set according to the part 1 of ISO 3601 standard.

HyperMesh v.14.0 program is used to mesh the structure, because the mesh size has a direct impact on the numerical results. Therefore, by reducing the mesh size from 0.4 to 0.05, the mesh convergence is studied in 0.05 steps. When the mesh size is less than 0.2, the numerical results have no obvious change. Therefore, the mesh size is 0.2, the number

of meshes for O-ring sealing structures are 4985, and the number of meshes for Butterfly-ring sealing structures are 7529. The APDL statement of ANSYS v.16.2 program is used to complete the parameter definition and solution. In the 2D axisymmetric analysis, plane 182 element is used. The material model of seal ring is the Mooney–Rivlin Hyperelastic with two parameters, and the material parameters are 1.84 and 0.47, which have been discussed in Section 2.1. The material of piston rod and groove is STS 430 steel (Elastic modulus 210 GPa, Poisson ratio 0.3). Loads applied and boundary conditions are the variable in this study, in which the pre-compression ratio is in the range of 8% to 16%, and the hydraulic pressure is in the range of 0 to 20 MPa. The contact pressure of groove side is the major cause of hydraulic leakage during assemble process, in order to enhance the contact peak pressure, the butterfly-ring is designed to elevate the contact pressure of groove side as Figure 2b shown. The maximum section diameter of butterfly-ring is the same as that of O-ring, which is 7 mm. Specific parameters are illustrated in the Appendices A–C.

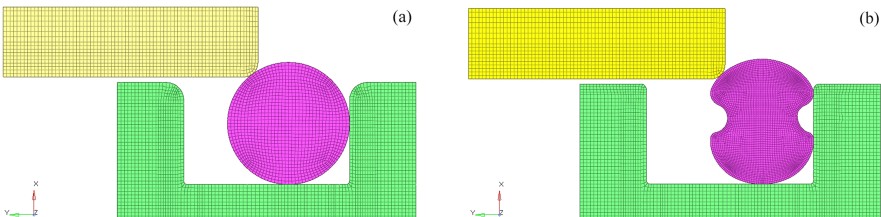

**Figure 2.** Finite element model. The original structure of seal is O-ring in (**a**), and the proposed structure of seal is butterfly-ring in (**b**), with the same maximum diameter as the O-ring. Both are assembled with groove in green mesh and piston rod in yellow mesh.

The contact pair, which is adopted to simulate the contact nonlinear behavior [24–26], is defined at the squeeze area during the installation mode and working mode, as shown in Figure 3. And the element type of TARGE169 and CONTA171 are adopted to simulate the contact behavior. The rubber ring is set as contact surface [27], due to its stiffness being softer than the material of piston rod and groove (Elastic modulus 210 GPa). Augmented Lagrange method, which could lead to better conditioning than the pure penalty method, is adopted to solve the contact problem of stiffness hardening for large displacement and deformation [28,29]. The material of the groove and piston rod is STS 430 steel, the surface roughness of the piston rod is 0.05 to 0.1 μm, and the surface roughness of the groove is 0.2 to 0.3 μm. Considering that the surface roughness of piston rod is less than that of groove surface, the friction coefficient between piston rod and sealing ring is set as 0.1, and the friction coefficient between groove and sealing ring is set as 0.2.

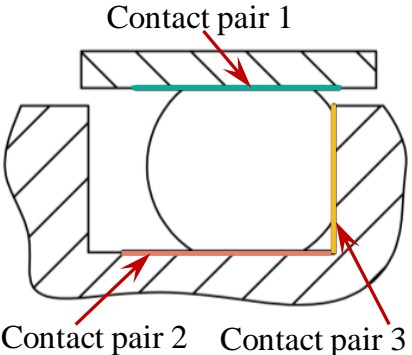

**Figure 3.** Modeling of contact pair. Three contact pairs are modeled, contact pair 1 is between piston rod and rubber ring, contact pair 2 is between groove side and rubber ring, contact pair 3 is between groove bottom and rubber ring.

### 3.2. Installation Mode

The boundary condition of FEA for the rubber ring is classified into two stage by installation mode and working mode. In order to study the deformation-stress of the rubber ring caused by boundary condition of assemble status, three kind of installation modes are assigned according to the Reference [30]. As schematic Figure 4 shown, piston rod axial displacement (RA), piston rod lateral displacement (RL), and groove lateral displacement (GL) are taken into account[15]. The conventional installation modes are presented in Figure 4b,c, e.g., piston rod is set with a rigid displacement in lateral direction and groove is set with a fixed constraint considered installation mode of RL. The RL and GL are considered nearly the same based on the theory of relative displacement. The detailed simulation parameters of installation mode are shown in Table A1 of the Appendix A. Additionally, piston rod is set with a rigid displacement in axial direction and groove is set with a fixed constraint considered installation mode of RA. The detailed simulation parameters are shown in Tables A2 and Table A3 of the Appendices A and B.

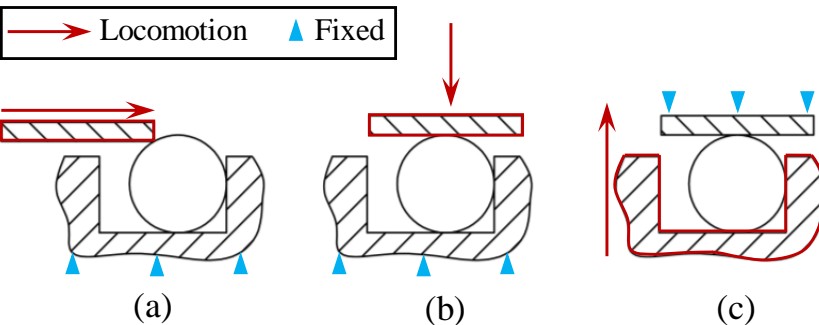

**Figure 4.** Modeling of installation mode. Three installation modes are used to simulate the process of pre-compression in public literature. RA mode means that the piston rod is applied with axial displacement and the groove is constraint with fixed displacement in (**a**). And RL mode means that the piston rod is applied with lateral displacement and the groove is fixed still as (**b**). The mode of GL means that the piston rod is fixed, and the groove is applied with lateral displacement in (**c**).

### 3.3. Working Mode

The boundary condition of working mode is set as static state with the constraint of piston rod and groove, hydraulic pressure on the rubber ring, shown in Figure 5.

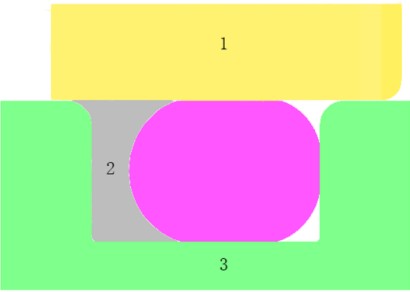

**Figure 5.** Modeling of working mode. The pressure of hydraulic fluid, which is represented with number 2, is applied to the deformed surface of the rubber ring by pre-compression function. And the piston rod numbered 1 and the groove numbered 3 are applied with fixed constraint.

It should be noted that the applied area of hydraulic pressure depends on the deformation of the rubber ring after the installation mode. Therefore, after the completion of installation simulation, the hydraulic pressure should be applied according to the hydraulic pressure action region in Figure 5, and then the simulation of working mode should be executed [31–33]. Additionally, the hydraulic pressure applied on the rubber ring is

differed in different experiment within the range of 0 to 20 MPa, as illustrated in Table A4 of the Appendix C.

## 4. Results

The influence of installation mode, installation stroke, pre-compression rate and hydraulic pressure on the sealing performance of the rubber ring are studied, 16 projects totally 154 experiments are conducted with numerical method [34]. Specific parameters are illustrated in Appendix. The value and corresponding code of variables are given in Table 1.

**Table 1.** Value and code of variable.

| Variable | Number | Series | | | | | | | | | | |
|---|---|---|---|---|---|---|---|---|---|---|---|---|
| Section diameter | Code | 1 | 2 | 3 | 4 | 5 | | | | | | |
| | Value (mm) | 1.8 | 2.7 | 3.6 | 5.3 | 7 | | | | | | |
| Pre-compression | Code | A | B | C | D | E | F | G | H | K | | |
| | Value (%) | 8 | 9 | 10 | 11 | 12 | 13 | 14 | 15 | 16 | | |
| Hydraulic pressure | Code | M | N | O | P | Q | R | S | T | U | V | W |
| | Value (MPa) | 0 | 2 | 4 | 6 | 8 | 10 | 12 | 14 | 16 | 18 | 20 |

### 4.1. Influence of Installation Mode

As Figure 6 shown, the contact length under RL mode is compatible to that under GL mode at piston rod side. When the pre-compression rate is 8%, 12% and 16%, the contact length under RA mode is 12%, 16% and 16% longer than that under RL mode, respectively. And the regular of contact length at groove side is compatible with the regular of contact length curve at piston rod side. Furthermore, the difference of contact length at groove side is smaller than that at piston rod side between RA and RL. Additionally, the difference of contact length between RA mode and RL mode is smaller at the groove side than at the piston rod side.

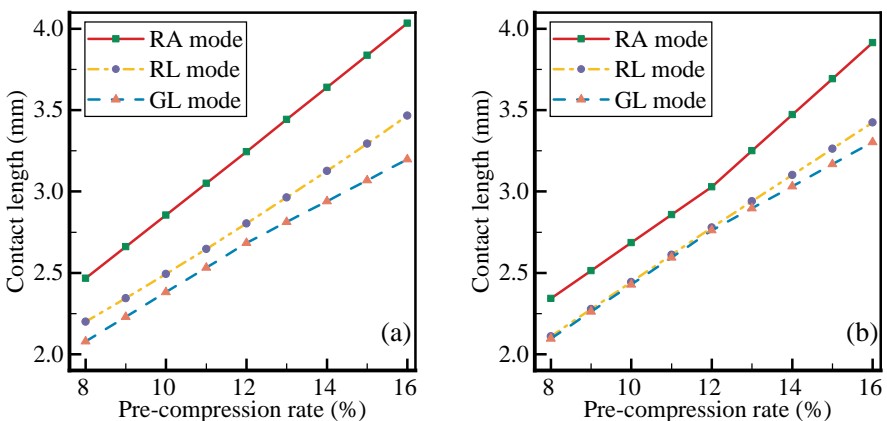

**Figure 6.** Comparison of installation mode on contact length. The contact lengths of the three installation modes are shown in (**a**) for piston rod side and in (**b**) for groove side, respectively. The three installation modes, including piston rod with axial displacement (RA mode), piston rod with lateral displacement (RL mode), and groove with lateral displacement (GL mode). The range of pre-compression rate is set between the maximum and minimum allowable value.

As Figure 7 shown, the regular of contact peak pressure at groove side is much compatible with that at piston rod side. Also, the regular of contact peak pressure at piston rod side and groove side is just like that of contact length. The contact peak pressure under RA mode is higher than that under RL mode and GL mode at piston rod side in Figure 7a. The smallest increase rate is 20% compared with RL under pre-compression rate 8%.

Only a half of the rubber ring is compressed with installation stroke 3 mm, thus the experiment 5EM3 is disqualified. As shown in Figure 8a, with the increase of installation

stroke, the increase of contact length at piston rod side, groove side and bottom sides become slower. When the installation stroke is increased to 9 mm, the limit value is reached. As shown in Figure 8b, with the increase of installation stroke, the contact peak pressure decreases to the limit value. When the installation stroke is increased to 9 mm, the limit value is reached. Additionally, the limit values of the contact peak pressure and the contact length are both obtained with the installation stroke of 9 mm.

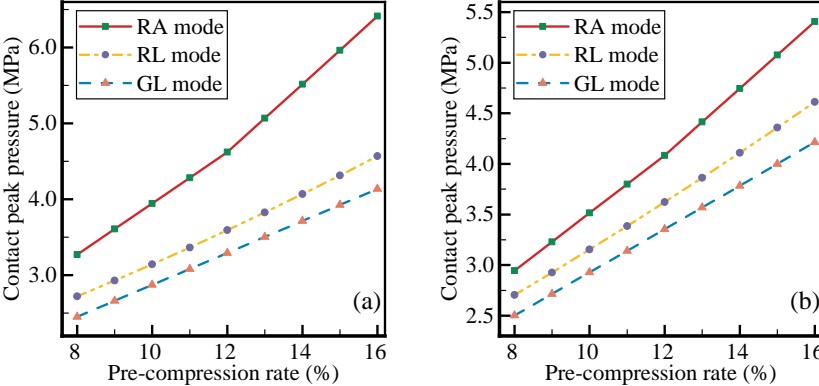

**Figure 7.** Comparison of installation mode on contact peak pressure. The contact peak pressure of the three installation modes are shown in (**a**) for piston rod side and in (**b**) for groove side, respectively.

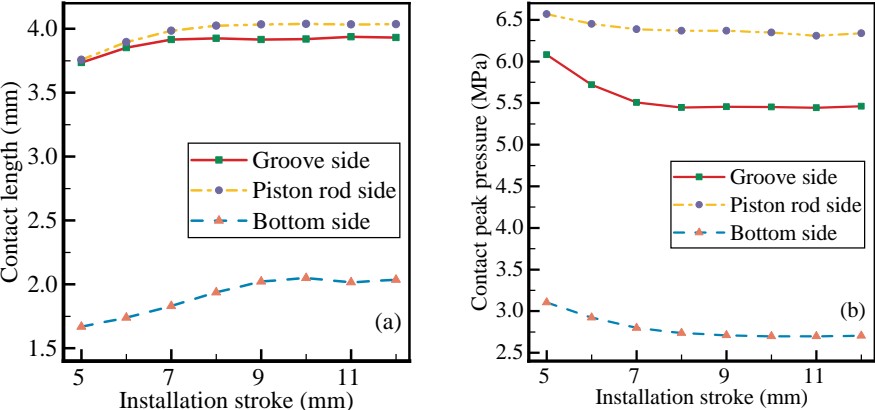

**Figure 8.** Comparison of installation stroke. The contact length in (**a**) and contact peak pressure in (**b**) under eight different installation strokes are compared at three positions, including groove side, piston rod side and bottom side.

### 4.2. Influence of Pre-Compression Rate

As shown in Figure 9a, the contact peak pressure at piston rod side increases linearly with the pre-compression rate, which can be fitted by linear fitting,

$$y = 0.45 + 0.31x \tag{8}$$

where $y$ is contact peak pressure, $x$ is the corresponding pre-compression rate. The linear formula is fitted with Pearson's 0.92. And the variation range of contact peak pressure is reflected by the bandwidth, which is in the range of 0.65 MPa to 1.69 MPa.

As shown in Figure 9b, the contact peak pressure at groove side is concentrated in a band for experiment 1M to 5M, also fitted by a linear formula,

$$y = 0.59 + 0.28x \tag{9}$$

The Pearson's is 0.97 for the fitted linear formula of contact peak pressure at groove side. The band is narrowed within the range of 0.29 MPa to 0.67 MPa.

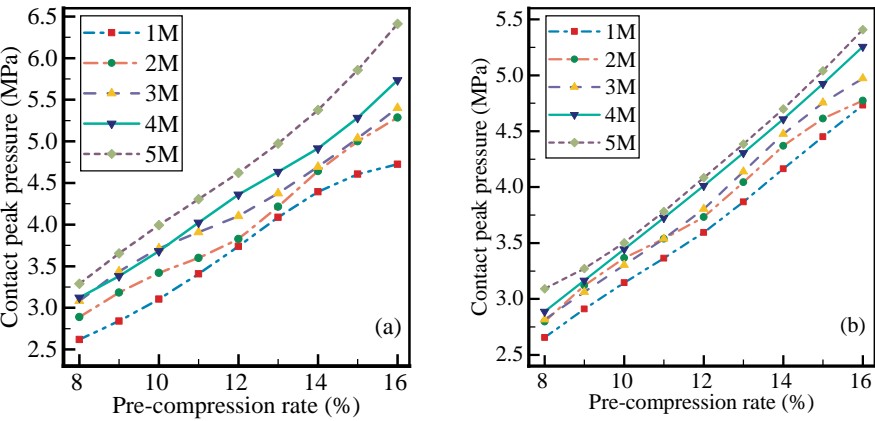

**Figure 9.** Comparison of pre-compression rate. The contact peak pressure under nine different pre-compression rates are shown in (**a**) for piston rod side and in (**b**) for groove side. Five groups simulation are compared, with five sealing rings of different section diameters coded from 1 to 5. And M represents for zero hydraulic pressure.

### 4.3. Influence of Hydraulic Pressure

According to the criterion that contact peak pressure should be higher than the hydraulic pressure, the failure zone of seal is defined. As shown in Figure 10a, the contact peak pressures of experiment from 5A to 5E are in the narrow band range of 2.10 MPa to 3.46 MPa. And in the range of hydraulic pressure from 0 to 20 MPa, the contact peak pressures are always distributed outside the failure zone. The relationship between contact peak pressure and hydraulic pressure is fitted as follows:

$$y = 4.71 + 0.99x \tag{10}$$

where $y$ is the contact peak pressure, $x$ is the hydraulic pressure. Also, the similar relation is fitted for grove side,

$$y = 3.90 + 1.20x \tag{11}$$

For the piston rod side and groove side, the relationship between the contact peak pressure and the hydraulic pressure are both conform to Pearson's 0.99.

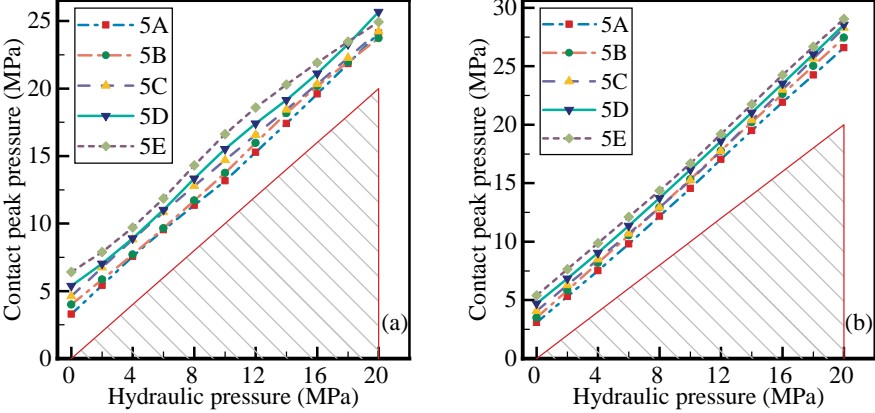

**Figure 10.** Comparison of hydraulic pressure. The contact peak pressure under different hydraulic pressures are shown in (**a**) for piston rod side and in (**b**) for groove side. Five groups simulation are compared, with five different pre-compression rates coded from A to E, and code 5 represents for the section diameter of the rubber ring (value is 7 mm). And failure zone is plotted to evaluate the sealing performance.

*4.4. Sealing Performance of Butterfly-Ring*

To compare the sealing performance of butterfly-ring and O-ring, the sealing simulations are carried out under assembly mode (hydraulic pressure 0 MPa) and working mode (hydraulic pressure 5 MPa). The maximum section diameter of the two types of seals is 7 mm, and the RA installation mode is adopted with the installation stroke of 9 mm. Considering that the pre-compression rate is the main factor affecting the sealing performance of the rubber ring, the pre-compression rates from 8% to 16%. Therefore, the experiment of butterfly-ring performance is composed of 18 simulation experiments of 2 groups.

The rubber rings at the piston rod side and at groove side are required to have good sealing performance, especially at the groove side which is used to be found leakage in hydraulic buffer of articulated landing gear. The sealing performance is compared under installation mode and working mode as Figure 11 shown.

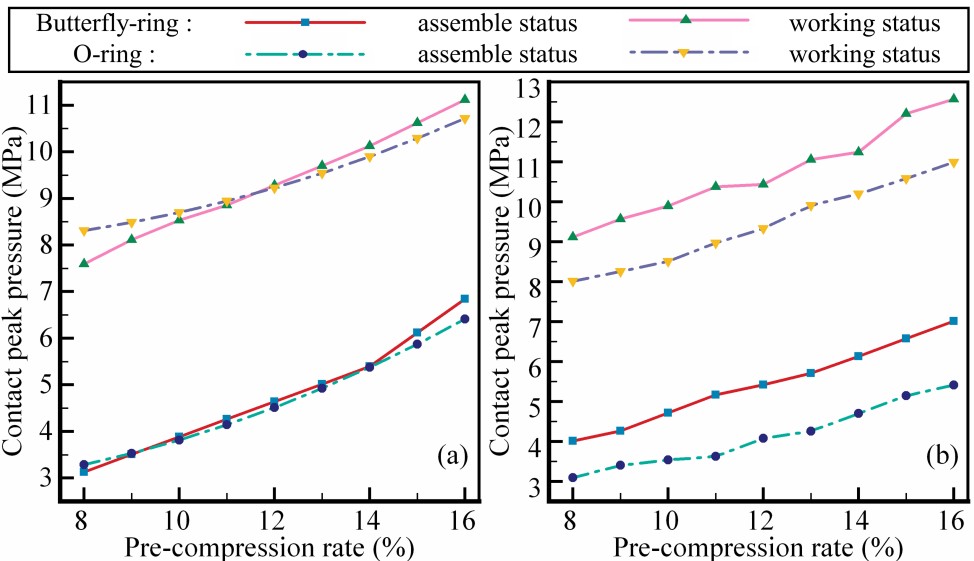

**Figure 11.** Verification of the proposed butterfly-ring. The comparisons of contact peak pressure between O-ring and butterfly-ring are made at piston rod side in (**a**) and at groove side in (**b**), with nine different pre-compression rates. Moreover, the contact peak pressure of O-ring and butterfly-ring under assemble status and working status are compared, respectively.

It can be found that the contact peak pressure of butterfly-ring is greater than that of O-ring at the maximum compression rate. When the pre-compression rate is 16%, the contact peak pressure at the groove side of butterfly-ring increases by 14%. And this law holds for both piston rod side and groove side. Moreover, it is worth noting that the proposed butterfly-ring improves the contact peak pressure at groove side and keeps the contact pressure at piston rod side close to that of the O-ring. Specifically, when the pre-compression rate is 8%, the contact peak pressure of butterfly-ring is 0.15 MPa lower than that of O-ring. When the pre-compression rate is 16%, the contact peak pressure of butterfly-ring is 0.45 MPa higher than that of O-ring. When the pre-compression rate is 14%, the contact peak pressure of butterfly-ring and O-ring is basically the same. As a whole, the contact peak pressure of butterfly-ring is close to that of O-ring at piston rod side, as shown in Figure 11a.

At the groove side, the contact peak pressure of butterfly-ring is always higher than that of O-ring, as shown in Figure 11b. Compared with O-ring, the contact peak pressure of butterfly-ring at groove side is increased about 30% under working status.

## 5. Discussion

### 5.1. Evaluation of Installation Mode

It can be drawn that the contact length and contact peak pressure caused by installation mode of RA are higher than that caused by installation mode of RL and GL. Considered the function between piston rod and rubber ring [35], the different interaction force for the RA, RL and GL installation mode are given in Figure 12.

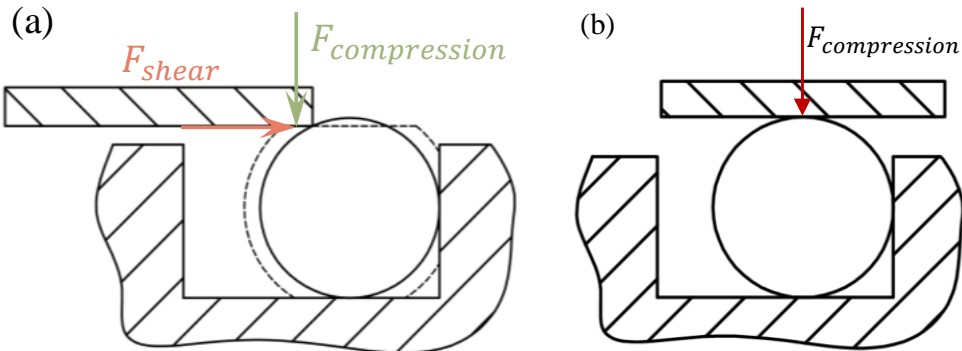

**Figure 12.** The force between piston rod and rubber ring in the installation mode RA is shown in (**a**). In the installation mode RL and GL, the interaction force is the same, and is shown in (**b**).

The deformation of the rubber ring under RA mode is caused by shear force and squeeze force together. The deformation of the rubber ring under RL and GL is nearly the same, due to the deformation are both caused by squeeze force as illustrated in Figure 12b [36]. The deformation of the rubber ring is affected by the stiffness of rubber. For rubber material, the stiffness caused by squeeze force are different with that caused by shear force. P. B. Lindley's research shows that the compression stiffness is more sensitive to displacement than the shear stiffness, and the relationship between the compression stiffness and displacement is of the fifth power, while the relationship between the shear stiffness and displacement is of the first power.

Thus, the pressure caused by compression stiffness would be much higher than the pressure caused by shear stiffness. It is consistent with the previous results that the contact peak pressure under RA is slightly higher than the contact peak pressure under RL and GL. Due to the contact peak pressure and contact length caused by RA, RL and GL are different, considered the three-dimensional hydraulic buffer as shown in Figure 1, the RA installation mode is approximately to the actual assemble [37]. Therefore, RA installation mode is adopted as the installation mode for FEA of rubber ring.

In addition, the installation stroke should be considered in the simulation of the rubber ring. As shown in Figure 8, the contact peak pressure arrives at the limit value with installation stroke 9 mm. Additionally, the limit value of contact length for groove side, piston rod side, and bottom side are also got stable at the installation stroke 9 mm. From the perspective of load transmission, the deformation and stress of the rubber ring is mainly triggered by installation stroke via the squeeze state. The squeeze state is simulated with contact pair defined among piston rod, rubber ring, groove as Figure 3 presented. Then, the squeeze state is calculated via the contact pair with Augmented Lagrange. In this method, the stiffness of the contact pair is rebalanced after each iteration, so the rubber ring needs enough installation stroke to reach a stable state. Therefore, the installation stroke should be 9mm in this study [38]. For different structure of the rubber rings, the installation stroke should be obtained through trial calculation.

### 5.2. Evaluation of Pre-Compression Rate and Hydraulic Pressure

The results of pre-compression rate experiment show that simply increasing the section diameter of the rubber ring cannot effectively increase the contact peak pressure, and properly increasing the pre-compression rate is effective for increasing the contact peak

pressure. P. B. Lindley's research shows that the relationship between compression stiffness and pre-compression rate is approximately a fifth power, so the compression stiffness of different section diameters are approximately equal under the same pre-compression rate. As shown in Figure 9, the contact peak pressure of simulation from 1M to 5M is concentrated in a narrow band. And the first-degree polynomial is used to fit the function of pre-compression rate to contact peak pressure, and the Pearson's value is above 0.92. Obviously, the contact pressure increased by the pre-compression rate is higher than that by the section diameter.

The influence of hydraulic pressure on the contact peak pressure of the rubber ring is caused by the influence of compression stiffness. In the working mode, the compression stiffness of the rubber ring is affected by the pre-compression rate and the hydraulic pressure, so the compression stiffness of the rubber ring under the hydraulic pressure is larger than that under the pre-compression rate. Accordingly, the contact peak pressure increases with the increase of hydraulic pressure. This point is also reflected in the fitting formula at groove side. The increase rate of contact peak pressure by hydraulic pressure (1.2) is greater than that by pre-compression rate (0.28). Correspondingly, 0.99 to 0.31 at piston rod side.

Although increasing the hydraulic pressure can improve the sealing pressure, but in engineering applications, the amplitude of hydraulic pressure is determined by the working load, and it cannot always be maintained at high value. Therefore, increasing the pre-compression rate is still an effective way to improve the sealing pressure.

### 5.3. Evaluation of Butterfly-Ring on Sealing Performance

Compared with O-ring, the proposed butterfly-ring improves the contact peak pressure at groove side. Because the stiffness of the rubber ring varies with its geometry, and the pre-deformed butterfly-ring improves the compression stiffness and is prone to lateral deformation, so the contact peak pressure of the butterfly-ring is higher than that of the O-ring at groove side. This is also confirmed by the calculation results, as shown in Figure 11. Under installation mode, the contact peak pressure of the butterfly-ring is increased by 30% at groove side, and the contact peak pressure increases by 14–16% under working mode.

In addition, the increase of contact peak pressure of butterfly seal to load is lower than that of O-ring. It shows that the sensitivity of butterfly seal is less than that of O-ring.

### 6. Conclusions

To solve the insufficient sealing at the groove side of articulated landing gear buffer, the sealing performance of the proposed butterfly-ring is discussed. It is found that the pre-deformation structure of butterfly-ring improves the contact peak pressure at groove side, which is beneficial to improve the sealing performance of buffer. Also, it is pointed out that the installation mode of RA is suitable as the boundary condition for the numerical analysis of rubber ring sealing, and the installation stroke should be obtained from the extreme value of contact length and seal pressure curve. Furthermore, the linear law of contact peak pressure with pre-compression rate, and contact peak pressure with hydraulic pressure are fitted, which can be used to calculate the pre-compression rate under required conditions.

**Author Contributions:** L.Z. is responsible for the theoretical modeling, numerical analysis, and academic writing. X.W. provides the academic and experimental guidance. 'A novel structure of the rubber ring for hydraulic buffer seal based on numerical simulation'. All authors have read and agreed to the published version of the manuscript.

**Funding:** This research was funded by the National Natural Science Foundation of China (Grant No. 51805249).

**Conflicts of Interest:** The authors declare no conflict of interest.

## Abbreviations

The following abbreviations are used in this manuscript:

FEA          Finite element analysis
M-R         Mooney–Rivlin
NBR         Nitrile rubber
RA           Piston rod axial displacement
RL           Piston rod lateral displacement
GL           Groove lateral displacement
$W$            Strain–energy function
$I$             Invariant of the Cauchy–Green strain tensor
$I_1, I_2, I_3$     Three invariants of the Green deformation tensor
$\lambda_1, \lambda_2, \lambda_3$     Three principal stretches ratios
$\sigma^{ij}$          Cauchy stress tensor
$\sigma_1, \sigma_2, \sigma_3$     Three principal Cauchy stresses
$\nu$             Poisson ratio
$c_{01}$          First material parameter of M-R model
$c_{10}$          Second material parameter of M-R model

## Appendix A. Parameter and Design of Installation Mode Simulation

**Table A1.** Parameters of installation mode experiment.

| Installation Mode | Direction | Pre-Compression Rate (%) | | | | | | | | |
|:---:|:---:|:---:|:---:|:---:|:---:|:---:|:---:|:---:|:---:|:---:|
| RA | Negative of Y | | | | | | | | | |
| RL | Negative of X | 8 | 9 | 10 | 11 | 12 | 13 | 14 | 15 | 16 |
| GL | Positive of X | | | | | | | | | |

The difference of the three installation modes are the different objects to which the displacement boundary are applied, and the influence of different installation modes on hyperelastic material need to be further discussed. In order to set a precise boundary condition for finite element analysis, three typical installation modes, including RA, RL and GL, are compared and analyzed. In the simulation of installation mode, three kinds of pre-compression rates of the rubber ring with section diameter of 7 mm are selected according to the ISO 3601 standard. The installation simulation is carried out under the parameters shown in Table A1.

**Table A2.** Parameters and labels of installation stroke experiment.

| Project | Label | Basic Parameter | Installation Stroke (mm) |
|:---:|:---:|:---:|:---:|
| | 5EM3 | | 3 |
| | 5EM5 | | 5 |
| | 5EM6 | Section diameter: 7mm | 6 |
| | 5EM7 | | 7 |
| 5EM | 5EM8 | | 8 |
| | 5EM9 | | 9 |
| | 5EM10 | Pre-compression: 16% | 10 |
| | 5EM11 | | 11 |
| | 5EM12 | | 12 |

Since the master displacement of RA mode is the axial direction displacement of piston rod, the installation stroke should be long enough to ensure that the rubber ring is in a stable compression state. In order to determine the appropriate value of installation stroke, the section diameter and pre-compression ratio are designed according to the maximum allowable value of the buffer, and the parameters and labels are shown in Table A2.

**Appendix B. Parameter and Design of Pre-Compression Rate Simulation**

To analyze the influence of pre-compression rate on the sealing performance of the rubber ring, a series simulation are designed according to ISO 3601 standard. Considering that the influence of pre-compression rate on sealing performance of the rubber ring varies with the change of section diameter, and the range of section diameter is set from 1.8 mm to 7 mm. The maximum pre-compression rate is set as 16%, which is the maximum pre-compression rate of the rubber ring with section diameter from 1.8 mm to 7 mm. A total of 45 simulations have been carried out. As the regular of parameters and labels design are the same, only the parameters and labels of 1M simulation are given, as shown in Table A3.

**Table A3.** Parameters and labels of pre-compression rate experiment.

| Project | Label | Section Diameter | Pre-Compression Rate (%) |
|---------|-------|------------------|--------------------------|
| 1M | 1AM | 1.8 mm | 8 |
| | 1BM | | 9 |
| | 1CM | | 10 |
| | 1DM | | 11 |
| | 1EM | | 12 |
| | 1FM | | 13 |
| | 1GM | | 14 |
| | 1HM | | 15 |
| | 1KM | | 16 |

**Appendix C. Parameter and Design of Hydraulic Pressure Simulation**

**Table A4.** Parameters and Labels of Hydraulic Pressure Experiment.

| Project | Label | Pre-Compression Rate | Hydraulic Pressure (MPa) |
|---------|-------|----------------------|--------------------------|
| 5A | 5AM | 8% | 0 |
| | 5AN | | 2 |
| | 5AO | | 4 |
| | 5AP | | 6 |
| | 5AQ | | 8 |
| | 5AR | | 10 |
| | 5AS | | 12 |
| | 5AT | | 14 |
| | 5AU | | 16 |
| | 5AV | | 18 |
| | 5AW | | 20 |

To analyze the influence of hydraulic pressure on static sealing performance of the rubber ring, 55 simulations of 5 groups are carried out under working mode. According to the static sealing requirement of radial seal in ISO 3601, the limit hydraulic pressure is set to 20 MPa. Since the setting of parameters and labels is regularly, only the parameters and labels of 5A simulation are given in Table A4.

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
