# Peer review of "A Novel Structure of Rubber Ring for Hydraulic Buffer Seal Based on Numerical Simulation"

_applsci, doi:10.3390/app11052036_

Round 1
Reviewer 1 Report
The authors proposed new rubber sealing design to solve the concurring problem of insufficient static sealing at groove side in landing gear applications in aircrafts.The authors use FE modelling to prove that their new design (butterfly design) provides better performance in terms of sealing capability than traditionally used O-ring design. A hyper elastic model is used to simulate the environment of rubber sealing during application and the performance of the rubber is simulated. The authors study the influence of precompression rate and hydraulic pressure on the performance of the sealing. The authors claim that a 30% increase in assembly state and 14% in working state in terms of contact pressure performance. The new design has enhanced static pressure sealing capabilities and conclude that the design of the sealing ring can significantly affect the sealing pressure and hydraulic buffer.
The article is interesting. Authors should carefully study the comments and make improvements to the article step by step. After major changes can an article be considered for publication in the "Metals".
Line 85 to 95 combine as a paragraph and not as bullet points (recommended)
There are a lot of variables and formulas, the authors should create a nomenclature table stating all symbols used and their meaning, add the table/list at the end or the start of the manuscript
A lot of the math shown in the paper can be removed, only keep the main formulas which show the final purpose of the equation (for example remove equations 1-6 or some of them) unless you derived these formulas on your own then they can be kept. (recommended)
In Figure 2, the proposed seal, how did you decide the radius in the middle, is it based on certain analysis or is it decided by trial and error?
The authors are encouraged to make the images in the figures larger to increase their clarity
Where did the authors get the materials properties of the rubber material from?
What was the coefficient of friction used in the model and why this value was chose? Please clarify.
The FE model lacks many details such as mesh size, number of elements, elements used, mesh convergence study, loads applied and boundary conditions, material properties and so on. Please provide detailed information on the FE model since this is the main theme of this paper.
Author Response
Please see the attachment.Thank you very much for your comment.

Reviewer 2 Report
Title : “A novel structure of rubber ring for hydraulic buffer seal based on numerical simulation “
In this work the authors provide simulation to test a novel structure of rubber ring with butterfly geometry. The effects of pre-compression rate and hydraulic pressure on the static sealing performance of rubber ring are investigated. The authors claim that this structure could be better than the current used O-ring.
General comment: The general aim of this work may be eventually interesting. Nevertheless, several point should reworked to improve the quality and the impact of the manuscript. In particular, formulas should be checked for correctness and all the used quantities should be clearly stated within the manuscript in order to allow interested readers to fully understand the logic flow of the work. The Finite Element model should be clearly explained, clearly stating the quantity of elements used, the meshing procedure, the eventual oversimplifications due to the lowering of the model dimensions, the kind of analysis performed by authors, all the issues due to the convergence of non linear materials as the rubber. The main results are plotted by fitting only 3 or 4-5 points. These amount of information seems to be quite low and perhaps not enough to reproduce eventual non linearities. As a consequence, the authors are requested to increase (about 8-10 points) the number of points for all these plots. Last, but not least, all the presented results are due to simulations. Could authors provide some further information due to real performances of rubber rings (e.g., a couple prototypes of the new ring design) to support their results or these results are only speculative ? In this way, the impact of the manuscript could be considerably improved.
Some detailed comments:
Equations (3,4,5)
*) The authors should check Equations for correctness, and clearly state all the used quantities.
* ) Equation (5) seems to be referred to a compressible materials. How the authors would like to use this in case of total uncompressibility ? Please explain in detail how to overcome all the mathematical issues due to v=0.5.
*) Equation (7) : the derivation of this Equation from the previous theory is not clear. Please improve and explain in better way.
3.1. Finite element model
*) The authors should explain in detail: the amount and the kind of elements used, the meshing procedure, the eventual oversimplifications due to the lowering of the model dimensions, the kind of analysis performed, all the issues due to the convergence of non linear analysis and materials as the rubber.
4. Results
*) Figures 6-11: all figures are plotted starting from 3-5 points. As already said, this amount of information is too low to reproduce eventual non linearities. Thus authors are requested to increase the amount of points to about 8-10 for each plot.
All the figure legends should clearly explained both in the main text and within the figure captions. Perhaps within the current version of the manuscript some symbols are difficult to understand.
5. Discussions
*) Perhaps it should read “Discussion”. This sections should not contain formulas, which should better suited for the “methods” section. Therefore, all this information should be moved to the relevant section.
6. Conclusion
The "Conclusion" section should present in a concise form the main achievement of the work with respect to the current state of the art. However, also a pointed list should be avoided. Please rework accordingly.
Author Response

(The authors gave the same response as above.)

Round 2
Reviewer 1 Report
why you use mm to describe surface roughness, usually it should be in micro meters instead of mm, please check this
Authors have answered the previous questions
Author Response
Please see the attachment.Thanks again for your professional advice.

Reviewer 2 Report
The authors revised the manuscript. No further comments.
Author Response
We thank the reviewer for the positive evaluation on our work,and the professional advice.